# Genome-Wide Identification of LATERAL ORGAN BOUNDARIES DOMAIN (LBD) Transcription Factors and Screening of Salt Stress Candidates of *Rosa rugosa* Thunb

**DOI:** 10.3390/biology10100992

**Published:** 2021-10-01

**Authors:** Jianwen Wang, Weijie Zhang, Yufei Cheng, Liguo Feng

**Affiliations:** College of Horticulture and Plant Protection, Yangzhou University, Yangzhou 225009, China; jwwang@yzu.edu.cn (J.W.); b377lw@163.com (W.Z.); hupic_112@163.com (Y.C.)

**Keywords:** *Rosa rugosa*, LBD transcription factor, salt stress

## Abstract

**Simple Summary:**

The transcription factor family LBD were well-known as regulator of plant development. Several recent studies indicated LBD genes response to abiotic stresses and abscisic acid. The salt tolerant rose (Rosa rugosa) distribute wildly in coastal saline sands are ideal materials of exploring the salt -responsive LBD genes. We found 41 RrLBDs from genome of wild rose and classified them into Classes I and II. Interestingly, many plant hormone response sites and abiotic stress response sites were predicted in promoters of some RrLBDs. In them, five RrLBDs (RrLBD12c, RrLBD25, RrLBD39 and RrLBD40) were significantly induced or depressed by salt stress. We predicted the five genes as salt response candidate genes of wild rose and the sites in their promotors maybe pointcut for further study.

**Abstract:**

LATERAL ORGAN BOUNDARIES DOMAIN (LBD) transcription factors are regulators of lateral organ morphogenesis, boundary establishment, and secondary metabolism in plants. The responsive role of *LBD* gene family in plant abiotic stress is emerging, whereas its salt stress responsive mechanism in *Rosa* spp. is still unclear. The wild plant of *Rosa rugosa* Thunb., which exhibits strong salt tolerance to stress, is an ideal material to explore the salt-responsive *LBD* genes. In our study, we identified 41 *RrLBD* genes based on the *R. rugosa* genome. According to phylogenetic analysis, all *RrLBD* genes were categorized into Classes I and II with conserved domains and motifs. The cis-acting element prediction revealed that the promoter regions of most *RrLBD* genes contain defense and stress responsiveness and plant hormone response elements. Gene expression patterns under salt stress indicated that *RrLBD12c*, *RrLBD*25, *RrLBD*39, and *RrLBD40* may be potential regulators of salt stress signaling. Our analysis provides useful information on the evolution and development of *RrLBD* gene family and indicates that the candidate *RrLBD* genes are involved in salt stress signaling, laying a foundation for the exploration of the mechanism of *LBD* genes in regulating abiotic stress.

## 1. Introduction

Transcription factors (TFs) play key roles in plant functional regulatory maps by regulating target gene transcription [1,2]. The plant-specific TF family LATERAL ORGAN BOUNDARIES DOMAIN (LBD) was identified by the conserved DNA binding domain LATERAL ORGAN BOUNDARIES (LOB), which includes a zinc-finger-like motif composed of cysteine residues “CX2CX6CX3C,” a Gly-Ala-Ser (GAS) block, and a spiral coiled structure similar to leucine (Leu) zipper related to protein dimerization, “LX6LX3LX6L” [3]. In particular, the Class II LBD protein only contains the zinc-finger-like structure in the LOB domain [3,4]. The *LBD* gene families of numerous plant species have been identified genome-wide. A total of 43 *LBD* genes were first discovered in *Arabidopsis thaliana* [5]. In addition, 44 *LBD* genes were identified in the monocotyledonous plant *Zea mays* [6]. Exactly 131 *LBD* genes were identified in the dicotyledonous plant allotetraploid cotton variety *Gossypium hirsutum* [7]. In Rosaceae, 58 and 37 *LBD* genes were identified from *Malus pumila* and *Fragaria vesca*, respectively [8,9].

The *LBD* family, which plays an important role in the development of inflorescences, leaves, and lateral branches of plants, is also known as ASYMMETRIC LEAVES2 LIKE [5,10]. The first isolated *AtLBD* (*LBD* of *A. thaliana*), that is, *AtLOB*, regulates the early development of leaves by interaction with SHOOT MERISTEMLESS and BREVIPEDICELLUS proteins [3]. Meanwhile, AtLOB targets the promoter of *PHYB ACTIVATION TAGGED SUPPRESSOR*1 gene to inhibit the accumulation of brassinosteroids, thus limiting the growth of organ boundaries [11]. The *AtLOB* homologous gene of rice (*Oryza sativa*), *OsRA*2, is involved in regulating the panicle stem development and seed growth morphology [12]. The *AtLOB* homologous gene of *Zea mays*, *ramosa*2, is involved in regulating the development of flower organ [13]. *AtLBD*6, one of the earliest isolated *AtLBD*, is expressed in the paraxial region of cotyledon primordium [14,15,16]. The *AtLBD*6 homologous gene *indeterminate gametophyte*1 (*IG*1) of *Zea mays* is the key gene to regulating leaf adaxial–abaxial patterning and embryo sac development [17]. In rice, the homologous gene *OsAS*2 regulates bud differentiation and leaf development and controls rice spikelet development, whereas *OsIG*1 restricts female gametophyte proliferation to regulate ovule development [18,19]. On the other hand, *LBD* genes participate in plant root development. *AtLBD16*, *AtLBD18*, *AtLBD29*, and *AtLBD33* regulate the formation of lateral roots by acting downstream of auxin signal transduction pathway mediated by *AtARF7* and *AtARF19* [20,21]. In addition, the *AtLBD*16/*AtLBD*18-*AtARF*7/*AtARF*19 module regulates adventitious root (AR) production [22]. Two homologous genes of rice, namely, *crown rootless*1 (*crl*1) and *Adventitious rootless*1 (*OsARL1*), regulate the growth of underground parts and development of rice by auxin response. *crl*1 regulates the formation of crown roots (CRs), whereas *OsARL*1 participates in the formation of ARs [6,23]. The *Zea mays* homologous gene *RTCS-LIKE*, which is expressed at the root crown primordium, regulates the CR development and rooting of the embryo bud [24,25]. In addition, Class II *LBD* genes regulate plant metabolic processes such as nitrogen and anthocyanin metabolism [13,15]. *AtLBD*37, *AtLBD*38, and *AtLBD*39 control NO^3-^/N signal transduction and inhibit anthocyanin biosynthesis of *A. thaliana* [26,27]. *OsLBD*37 and *OsLBD*38 delay the heading date by depression of the flowering gene, thus regulating rice yield [28].

Although the conserved function of *LBD* genes in the regulation of the development of lateral organs has been proven in model plants and non-model species [9,29,30], the emerging role of *LBD* in plant abiotic stress has been indicated by recent studies. Several newly constructed genome-wide expression profiles identified that several *LBD* members can respond to abiotic stress or abscisic acid (ABA) treatment, e.g., *SbLBD32* of *Sorghum bicolor*, which is highly induced by salt and drought. *CsLOB_3* and *CsLBD36_2* of *Camellia sinensis* are induced by salt and drought and involved in improving the promoter activity of *CsC4H* (*cinnamate 4-hydroxylase*), *CsDFR* (*dihydroflavonol 4-reductase)*, and *CsUGT84A* (*UDP-glucuronosyltransferase*), which are key genes of the flavonoid biosynthesis pathway [31]. *PpLBD*27 of *Physcomitrella paten* responds to drought stress through the ABA signaling pathway [32].

*Rosa rugosa* is an important aromatic plant whose flowers are rich in terpene aromatic substances. Its commercial cultivars are widely used for spice industry materials or scented teas but belong to glycophyte, which lacks salt tolerance. The wild *R. rugosa* plants are distributed naturally in the coastal area of northeast China, Russian Far East, the Korean peninsula, and Japan. The plant evolved a strong salt tolerance to adapt to its growth environment (high salinity beach), making wild *R. rugosa* an excellent material for a salt tolerance study of *Rosa* genus plants. In the past several years, the whole-genome sequencing of horticultural plants revealed their long history of evolution and artificial domestication. The whole genome of rose at the chromosome level released this year (2021) has lain the foundation for its analysis and genetic transformation at the gene level. Since the release of genomes of *R. rugosa* and its sister species *R. chinensis* recently [33,34], *RrLBDs* (*LBDs* of *R. rugosa)* or *RcLBDs* (*LBDs* of *R. chinensis*), including the potential salt-responsive *RrLBD* members, have not been identified. This study aimed to screen the *RrLBD* genes involved in salt stress response by genome-wide phylogenetic analysis, chromosome location analysis, gene structure, and expression profile analysis. This study provides *LBD* gene candidates for further research on salt stress mechanism regulation in *R. rugosa* and will be helpful in the cultivation of salt-tolerant rose species by genetic engineering.

## 2. Materials and Methods

### 2.1. Identification of RrLBD Family

To identify all the members of *RrLBD*, we searched the hidden Markov model profile of the LOB domain (Pfam number PF03195) in the whole genome of *R. rugosa* (http://eplantftp.njau.edu.cn/Rosarugosa/, accessed on 15 May 2021) by HMMER 3.0 [35,36]. The hypothesized LOB domains of candidate *LBD* genes were checked in the Conserved Domain Database (CDD, https://www.ncbi.nlm.nih.gov/, accessed on 15 May 2021) and Pfam database (http://pfam.xfam.org/, accessed on 15 May 2021) [35,37]. The number of amino acids (AAs), molecular weight (MW), and isoelectric point (pI) of protein were obtained by ExPASy (https://web.expasy.org/protparam/, accessed on 15 May 2021) tool [38].

### 2.2. Phylogenetic Analysis of LBD Family

All the members of *RcLBD* family were identified from the genome of *R. chinensis* by the same method. *AtLBDs* were downloaded from the PlantTFDB (http://planttfdb.gao-lab.org/, accessed on 15 April 2021) and checked in accordance with a reference study [33]. A neighbor-joining (NJ) tree of protein sequences of *RrLBDs*, *AtLBDs*, and *RcLBDs* was built by MEGA-X (https://www.megasoftware.net/, accessed on 15 May 2021) with p-distance and pairwise deletion parameters [39,40]. A total of 1000 bootstrapping replications were used to verify the reliability of the phylogenies, and the online tool ITOL (https://itol.embl.de/, accessed on 15 July 2021) was used for coloring [41].

### 2.3. Genen Structure, Motif Composition, and Cis-Acting Elements Analysis of RrLBD Family

The motifs were predicted by the online tool MEME 5.3.3 (https://meme-suite.org/meme/, accessed on 15 May 2021) with default parameters. The cis-acting elements of *RrLBD* genes were predicted from promoter regions (2000 bp upstream the gene loci) by PlantCare [42]. The gene structures, motifs, and cis-acting elements were visualized by Tbtools 1.086 [43].

### 2.4. Synteny Analysis of LBD Genes

The inter-species synteny analysis was conducted by reciprocal BLASTP search for potential homologous gene pairs (E < 10^−5^, top five matches). The syntenic region and homologous pairs of *LBD* genes were illustrated by TBtools 1.086 [43].

### 2.5. Expression Analysis of RrLBDs in Different Tissues under Salt Stress

The wild shrubs of *R. rugosa*, which are naturally distributed in the coastal beach of Western hills north village of Muping district, Yantai city, Shandong province, China (37.455° N, 121.692° E), were selected as plant materials. In July 2019, three wild plants were dug out from the sand, and their roots were treated with 170 mM NaCl solution immediately for 1 h (ST). Another three plants that were soaked in deionized water for 1 h were set as the control group (CK). The leaves and roots of ST and CK plants (three replications for each) were picked and frozen by liquid nitrogen immediately and stored at −80 °C. The RNA for each sample was extracted by MiniBEST Universal RNA Extraction Kit (TaKaRa, Japan), and 12 libraries, i.e., three replications for the leaves of ST (L1h), roots of ST (R1h), leaves of CK (LCK), and roots of CK (RCK), were constructed for transcriptome sequencing (RNA-seq) on an Illumina NovaSeq 6000 platform. After filtering raw reads, the fragments per kilobase of exon model per million mapped fragments (FPKM) was calculated by mapping clean reads to the *R. rugosa* genome using HISAT 2.2.1 (http://daehwankimlab.github.io/hisat2/, accessed on 15 May 2021). The expression profile of *RrLBDs* was constructed with FPKM. Foldchanges of *RrLBDs* were calculated on the basis of the read reads using R package DESeq2 (http://www.bioconductor.org/packages/release/bioc/html/DESeq2.html, accessed on 15 May 2021).

Quantitative real-time polymerase chain reaction (qRT-PCR) was conducted using ChamQ SYBR Color qPCR Master Mix (Vazyme, China) on a CFX96 Real-time PCR platform (Bio-Rad, China). The PCR reaction system and program were conducted following the manufacturer’s instructions. Three biological replicates with three technical replicates were prepared for each sample [44]. Appendix A lists the primers of target genes and internal reference gene.

## 3. Results

### 3.1. Identification and Phylogenetic Analysis of RrLBD Family

A total of 41 RrLBD TFs were identified by PF03195 model, and their *LOB* domains were verified in the CDD and Pfam database. The gene names were numbered in accordance with the top one BLASTP search of known AtLBD TFs (Appendix A). In addition, homologous *RcLBDs* were distinguished by lowercase letter suffixes, and isoforms (putative alternative splicing *RcLBD* mRNA) were distinguished by number suffixes. This situation only existed in *RcLBD* proteins. The biochemical character of *RrLBD* proteins changed within a minimal range. A total of 41 *RrLBDs* proteins ranging from 144 AA (*RrLBD*24) to 386 (*RrLBD*27a) AA had a MW ranging from 16.1 kDa (*RrLBD*12a) to 44.6 kDa (*RrLBD*27a), and their pI changed from 4.99 (*RrLBD*7) to 8.91 (*RrLBD*2b) (Appendix A).

The phylogenetic tree (Figure 1) was constructed by the full-length protein sequences of 41 *RrLBDs*, 43 *RcLBDs*, and 43 *AtLBDs* belonging to two categories (Class I and II), referring to the known topological structure of *AtLBD* family [45,46]. Seven groups, namely, Ia, Ib, Ic, Id, and Ie of Class I and IIa and IIb of Class II, were divided on the basis of the sub-topological structure. Five homologous gene pairs (*LBD*37, 38, 39, 40, and 42) of three species, except the missing homologs of *AtBD41*, indicated that the LBD of Class II was highly conserved. In Class I, Ie, containing four *AtLBDs*, is a unique class found in *A. thaliana.* A total of 13, 3, 12, and 8 *RrLBDs* and 18, 4, 10, and 6 *RcLBDs* belong to Ia, Ib, Ic, and Id, respectively, which indicated that *RrLBDs* have a stronger congruent relationship with *AtLBDs* (10, 3, 12, and 8 for each sub-class).

### 3.2. Gene Structure and Motif Composition Analysis of RrLBD Family

On the basis of the NJ phylogeny of RrLBDs and RcLBDs (Figure 2A), the 10 significantly enriched MEME motifs indicated that the region of the top four motifs with 2, 3-1, and 4 arrangement (Figure 2B), which overlapped with the complete LOB domains, corresponded to the zinc-finger-like motif (CX2CX6CX3C, Figure 2D (I)), GAS block (Figure 2D (II)), and the spiral coiled structure similar to Leu zipper (LX6LX3LX6L, Figure 2D (III)), respectively. The four motifs were conserved in all Class I LBDs (Figure 2B,D)). Class II RrLBDs only contained motif 2 in the location of LOB domains. In addition, motifs 5 and 9 located at the C-terminal were motifs specific to Ia, whereas motifs 6 and 10 specific to Class II LOB domain replaced the location of motifs 3 and 4 of Class I LOB domains, respectively. Compared with AtLBD proteins, in addition to the important proline residue, another glycine residue in the motif 3 specific to RrLBD/RcLBD proteins is highly conserved, and it may be an important residue of the DNA binding domain. This finding showed that different AA residues in the LOB domain are indispensable to the characteristic function of family members, and the protein change in TFs will lead to the changes in its function of recognizing the promoter region [4,47].

The gene structures of *RrLBD/RcLBD* homolog pairs were highly consistent (Figure 3C). The variability degree of gene structure is related to the diversity of gene members belonging to the same group. Class I *RrLBDs* with exons ranging from 1 to 5 (1 to 4 exons for *RcLBDs*) include 7 intron-free genes, 25 genes with two exons, 3 genes with three exons, 1 gene with four exons, and 1 gene with five exons. Compared with the high gene structure diversity of Class I, Class II is relatively conservative. The coding sequence regions of most Class II *RrLBDs* (and *RcLBDs*) are split by a short intron, with the exception of *RrLBD*38, which has two more long introns.

### 3.3. Chromosomal Distribution and Evolutionary Analysis of RrLBD Genes

The chromosome localization map (Figure 3A) showed that *RrLBDs* were distributed dispersedly in five chromosomes (Chr1 and Ch4–7) and clustered in Chr2 and Ch3 with two gene clusters. The five paralogous gene pairs, namely, *RrLBD*2a/2b, *RrLBD*12a/12b, *RrLBD*16.1/16.2, *RrLBD*17.1/17.2, and *RrLBD*29.1/29.2, were located in these gene clusters.

The synteny analysis identified 38 *LBD* orthologous gene pairs of *R. rugosa* and *R. chinensis* (Figure 3B) and 20 pairs of *R. rugosa* and *A. thaliana* (Figure 3C). More pairs of orthologous *RrLBD/RcLBD* were in accordance with the LBD proteins phylogeny of three species (Figure 1), and this finding indicated that the lineage division of *LBD* genes was smaller between rosaceous plants after separation from the ancestor of three species. In particular, *RrLBD*37 and *RrLBD*38 of Chr2 and *RrLBD*39 and *RrLBD*40 of Chr6 were homologous to at least two *RcLBD* genes, indicating that Class II members may have functional differentiation between *R. rugusa* and *R. chinensis*.

### 3.4. Analysis of Cis-Acting Elements of RrLBD Promotors

A total of 19 cis-acting elements in *RrLBD* promoter regions were identified. These cis-acting elements were classified into five categories: plant hormone response, light response, stress response, specific binding site of MYB, and endosperm expression (Figure 4). First, the plant hormone response-related category contained the most elements, including methyl jasmonate (MeJA) responsiveness (CGTCA motif), ABA responsiveness (ABRE), salicylic acid responsiveness (TCA element), gibberellin- responsive element (GARE motif), gibberellin-responsive element (TATC box), and auxin-responsive elements (TGA element) or a part of an auxin-responsive element (TGA box). ABRE is the most widely distributed with 82 sites, whereas TGA box is distributed in 3 sites. The light response-related category contained the second highest number of elements, including light-responsive element (GT1 motif), light responsiveness (G box), a part of a module for light response (AE box), and a part of a light response element (CAG motif). Meanwhile, the stress response category including four elements, namely, wound-responsive element (WUN motif), defense and stress responsiveness (TC-rich repeats), low-temperature responsiveness, and enhancer-like elements, are involved in anoxic-specific inducibility (GC motif). The specific binding site of MYB category included the MYB binding sites (MBS) involved in drought inducibility, flavonoid biosynthetic gene regulation (MBSI), and light responsiveness (MRE). The last category included the endosperm expression element (GCN4 motif) with three sites in all promotors. These predicted binding sites or elements indicated that *RrLBD**s* may be targeted by related TFs involved in response to plant hormone, light, abiotic stress, and endosperm development.

### 3.5. Salt-Responsive Expression Analysis of RrLBDs

To screen the salt stress-responsive *RrLBD* candidates, we screened eight *RrLBDs* (with one isoforms) belonging to differentially expressed genes out from roots (R1H versus RCK) and leaves (L1H versus LCK) by RNA-seq data. The expression profile based on normalized FPKM values (Figure 5A, Appendix A) manifested that *RrLBD*40, *RrLBD*25, *RrLBD*19, *RrLBD*12c, and *RrLBD*4.1 in roots (R1H) and *RrLBD*40, *RrLBD*39, and *RrLBD*38 in leaves (L1H) were induced significantly by salt. The abundance of five *RrLBDs* in leaves were extremely low for accurate detection (FKPM < 1), and this finding indicated that most *RrLBDs* especially respond to salt in roots. The relative expression level determined in qRT-PCR (Figure 5B) proved that the profile based on transcriptome data was credible and indicated the statistical significance of four salt-responsive candidates. *RrLBD*40 was significantly induced in roots and leaves, *RrLBD*39 was significantly induced only in leaves, and *RrLBD*12c was significantly induced only in roots. In particular, as the only member significantly depressed by salt in roots, *RrLBD*25 can be a negative regulator of salt response by targeting salt-sensitive genes. The four genes can be candidates of strong response to salt stress.

## 4. Discussion

### 4.1. High Conservation of RrLBD Family

A total of 41 *RrLBDs* were located on all the seven chromosomes of *R. rugosa* genome. The gene number is similar to that of *R. chinensis*, *A. thaliana*, *Z. mays*, *Solanum lycopersicum*, and *M. pumila*. Thus, the gene number of the *LBD* family is highly conservative with minimal interference from ancient polyploidization events involving diploid angiosperms [48]. Several paralogous gene pairs located on *RrLBD* clusters of Chr2 and Chr3 indicated that these paralogous *RrLBDs* should evolve from tandem replication. In addition to the stable gene number, the lineage of the *RrLBD* family (two groups including seven subgroups) was consistent with the classification of *LBD* genes in other species. The homologous pairs clustered in the same branch generally have conserved LOB domains whose roles may be similar to those of the corresponding well-studied *AtLBD**s* [49].

### 4.2. Salt Response of RrLBD Candidates

Soil salinity is one of the main environmental stress factors affecting plant growth and development [50]. In the past decade, TFs, including *ERF*, *MYB*, *WRKY*, *NAC*, and *bHLH*, have been proven to respond to salt and regulate downstream response gene expression [51,52]. The transcriptional regulation mechanism of the *LBD* gene family under salt stress is still unclear. Most *RrLBD* genes express highly in specific tissues. Except *RrLBD*39 and *RrLBD*12c, *RrLBD*25, *RrLBD*4.1, *RrLBD*4.2, *RrLBD*13, *RrLBD*19, *RrLBD*38, and *RrLBD*40 (Figure 5A) are significantly more abundant in roots. These tissue-specific *RrLBDs* are up- or downregulated in roots and/or leaves under salt stress, and several genes should be salt-responsive *RrLBD* candidates.

First, for the significant depression in roots and leaves under salt stress, *RrLBD*25 may negatively respond to salt stress signal. Further, the defense and stress responsiveness elements in the promoter region of *RrLBD*25 and its homologous gene *AtLBD*25, which plays a transcriptional role in auxin signaling, indicate that *RrLBD*25 may respond to salt by cross talk with auxin signaling [53]. An abiotic stress response study of auxin-related gene families in *S. bicolor* indicated that *SbLBD*32 was highly induced by salt and IAA. This result implies that pathways cross talked between auxin and abiotic stress signaling.

The expression of *RrLBD*39 in leaves and *RrLBD*40 in roots changed acutely under salt stress. In addition to the highly induced abundance in salt-treated leaves, two MeJA responsiveness elements (CGTCA motif) of the *RrLBD*39 promoter region indicated that *RrLBD*39 is a positive regulator of salt. Abiotic stresses, such as drought and salt stresses, can promote the production of secondary metabolites in plants [54,55]. Class IIb gene *RrLBD*39 is homologous to *AtLBD*37, *AtLBD*38, and *AtLBD*39, regulating anthocyanin synthesis and nitrogen metabolism [26]. In *Solanum tuberosum*, *StLBD*1-5 is the homologous gene pair of *AtLBD*39, and its expression is downregulated under drought stress. This condition is due to the low expression of *StLBD*1-5, which promotes the accumulation of anthocyanins in leaves, thus improving the plant’s drought resistance [56]. The salt response and potential metabolism regulation role of *RrLBD*39 indicate the balance or promoting relationship of salt tolerance and metabolism of *R. rugosa.* Moreover, MeJA is a kind of plant hormone that regulates defense mechanism and stress response [57]. MeJA can promote the transcription modification of several genes, thus participating in the resistance to salt stress by promoting the secondary metabolism of plants [58]. In *C. sinensis*, the expression analysis of *CsLBDs* after MeJA treatment showed that the expressions of *CsLBD*38 and *CsLBD39_2* were upregulated, and both belong to the Class II subgroup. Combined with the research evidence of this experiment, *RrLBD*39 may be connected in series with the MeJA signaling pathway to participate in plant secondary metabolism and defense response to salt stress as hormone mediation. Finally, unlike the homologous gene of *AtLBD*41 regulating the cell specialization of the paraxial region [59], *RrLBD*40 has a large number of cis-acting elements involved in ABA responsiveness. ABA is an important plant stress hormone, and it plays an important role in salt stress signaling [60]. ABA in plants can interact with MYB TFs and play a role as a negative regulator of salt stress [61]. In addition, several transcriptomics studies showed that most genes of the ABA synthesis and signaling pathway are up- or downregulated under salt stress, and ABA signal pathway is involved in the regulation of salt stress-related genes [62,63]. ABA signaling pathway can also increase the activity and expression of ion transporters, thus regulating the ion homeostasis of plants [64]. The nearly 10 times increased expression of *AtLBD40* in salt-treated roots may be amplified by the ABA signaling pathway to actively respond to salt stress signals. In addition, different binding sites of *MYB* TFs were observed in the promoter regions of *RrLBD*39 and *RrLBD*40. Thus, *RrLBD*39 and *RrLBD*40 may be target genes of *MYB* TFs. In *Salvia miltiorrhiza*, SmLBD50 interacts with SmMYB36/97 protein and participates in jasmonate signal transduction [65]. Therefore, on the basis of previous studies and this experiment, we find that *RrLBDs* may have a complex molecular regulatory network with other TFs, and this association is important for abiotic stress resistance, growth and development, secondary metabolism, and other important plant biological processes.

According to collinearity analysis, similar to *RrLBD*25 and *AtLBD*25, *RrLBD*4.1 and *AtLBD*4, *RrLBD*4.2 and *AtLBD*3, and *RrLBD*19 and *AtLBD*19 are homologous gene pairs. *PtLBD*4 is involved in the regulation of secondary growth in poplar, whereas *AtLBD*19 regulates callus formation; however, the specific function of *LBD*3 has not been clarified [66,67]. Other candidate genes, such as *RrLBD*12 and *RrLBD*13, need to be studied to determine their corresponding salt stress mechanisms.

## 5. Conclusions

In this study, we identified 41 *RrLBD* genes from the *R. rugosa* genome by bioinformatics. According to phylogenetic analysis, all *RrLBD* genes were divided into seven subclasses of two classes. The conserved motifs, gene structures, cis-acting elements, and collinearity analysis showed the conservation of the *RrLBD* gene family. The expression profiles of roots and leaves under salt stress indicated that *RrLBD*25, *RrLBD*4.1, *RrLBD*4.2, *RrLBD*12c, *RrLBD*13, *RrLBD*19, *RrLBD*38, *RrLBD*39, and *RrLBD*40 are related to salt stress. *RrLBD*39 and *RrLBD*40 were selected to be important salt stress-responsive candidates.

## Figures and Tables

**Figure 1 biology-10-00992-f001:**
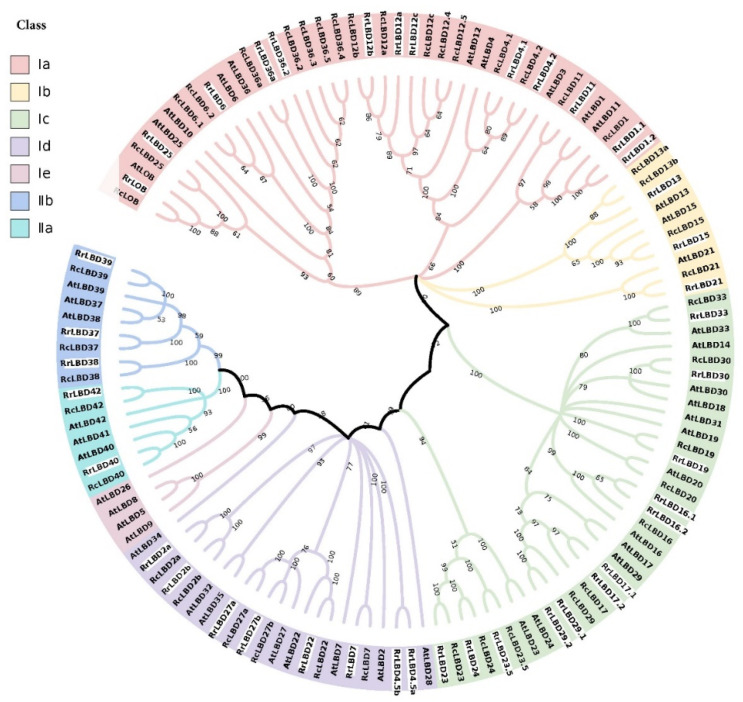
The NJ phylogeny of LBD transcription factors of *Rosa rugosa* (RrLBD), *Rosa chinensis a* (RcLBD), and *Arabidopsis thaliana a* (AtLBD). Ia-Ie of Class I and IIa-IIb of Class II are distinguished by pale pink, yellow, green, purple, dark pink, cyan, and blue, and RrLBDs are highlighted by the white background. Numbers on branches are bootstrap values.

**Figure 2 biology-10-00992-f002:**
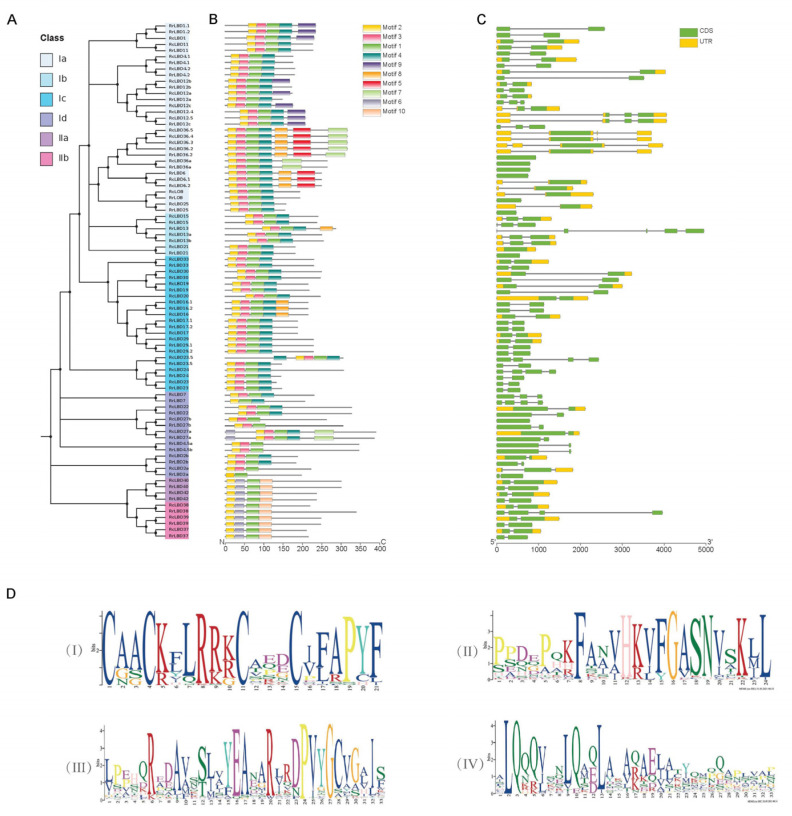
The NJ-tree (**A**), motifs (**B**), and gene structures (**C**) of LBD transcription factors of *Rosa rugosa* and *Rosa chinensis*. Classes Ia-d, IIa, and IIb are distinguished by nodes of different colors (**A**). The coloring boxes represent motifs whose locations are labeled by scaleplate of amino acid residue (**B**). The exons (rectangles) separated by introns (lines) are colored with blue (coding sequence, CDS) and yellow (untranslated region, UTR) whose locations are labeled by the nucleotide scaleplate (**C**). The top 4 significant enriched motifs are listed as sequences logos (**D**).

**Figure 3 biology-10-00992-f003:**
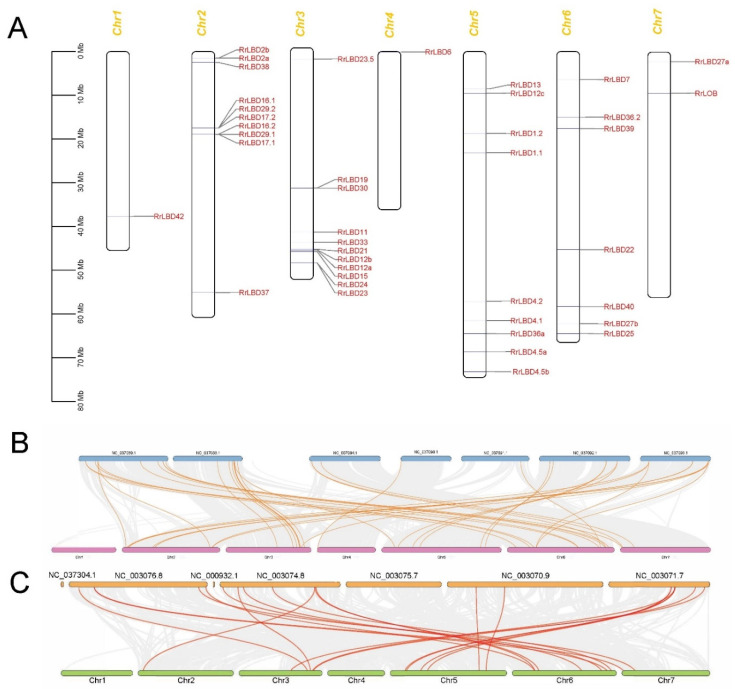
The chromosome location of *RrLBDs* (**A**) and intra-species synteny analysis of *Rosa rugosa* genome with *R. chinensis* (**B**) and *Arabidopsis thaliana* (**C**). Gene loci are mapped in chromosomes (**A**) labelled by the scaleplate of million base pairs (Mb). The gray lines indicate the collinearity block between the two genomes, and homologous *LBD* gene pairs are highlighted by red lines (**B**,**C**). The chromosomes ID of *Rosa rugosa* are labelled by Chr1-7. The chromosomes ID of *R. chinensis* and *A. thaliana* are labelled by accession number of NCBI genome database.

**Figure 4 biology-10-00992-f004:**
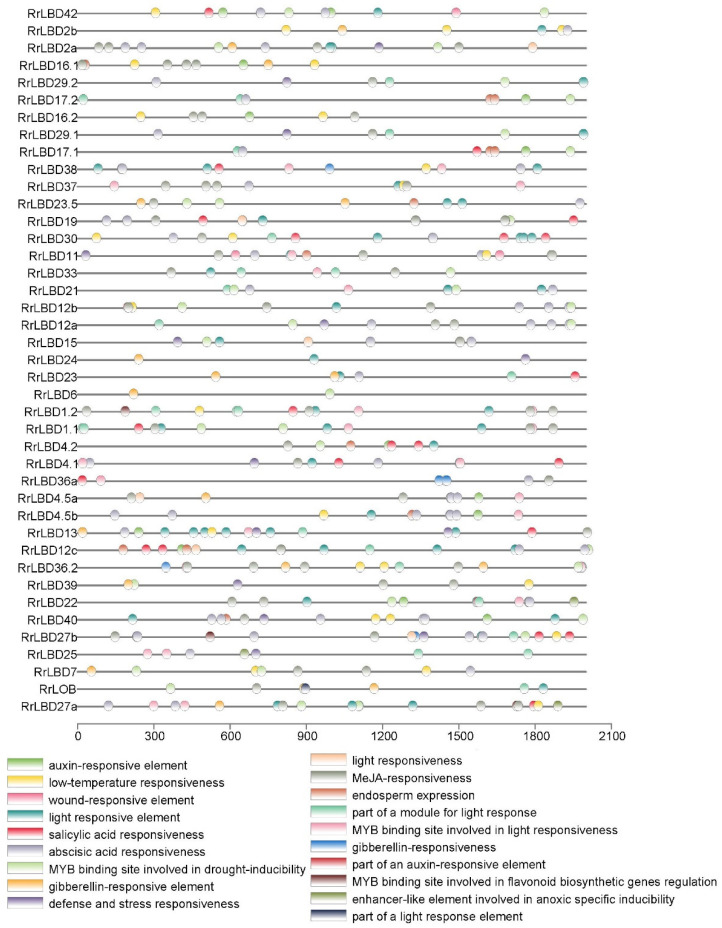
The cis-acting elements in promotors of *RrLBD*. A total of 19 predicted cis-acting elements (colored ovals) were located in promoter region 2000bp upstream of coding sequence (nucleotide number scaleplate).

**Figure 5 biology-10-00992-f005:**
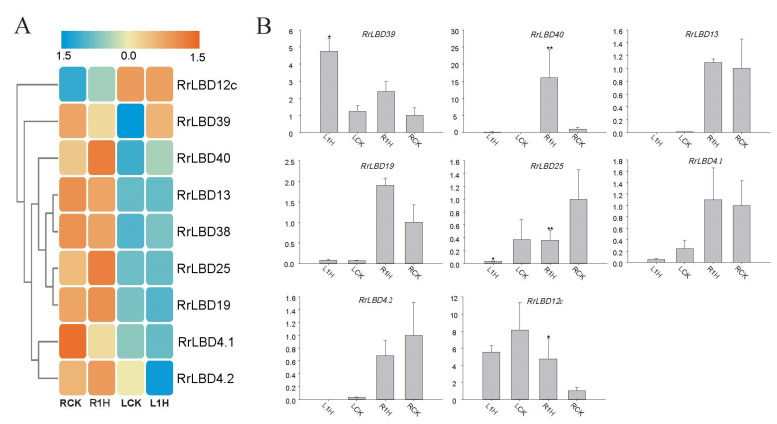
The differential expressed *RrLBDs* in roots and leaves of *Rosa rugosa*. The heatmap (**A**) was based on Log2 (FPKM) by the orange-blue gradation. L1h and R1h represent the leaves and roots of salt stress treatment, while LCK and RCK represent the control groups of leaves and roots. *RrLBD* genes are represented by Euclidean distance completely linked clustering. *RrLBD* expression level validation of real-time quantified PCR (**B**). RCK was set as the reference of relative expression level. * and ** indicate significance of 0.05 and 0.01 by two-sample *t*-test, respectively.

## Data Availability

Raw data of RNA-seq was deposited in SRA database (PRJNA752934).

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
