# Peer review of "Genome-Wide Identification of LATERAL ORGAN BOUNDARIES DOMAIN (LBD) Transcription Factors and Screening of Salt Stress Candidates of *Rosa rugosa* Thunb"

_biology, 2021, doi:10.3390/biology10100992_

Round 1
Reviewer 1 Report
This MS reports the whole-genome wide analyses on the LBD genes in Rosa rugosa. The authors performed structural and expressional analyses. I found that these analyses are very comprehensive and informative. I am pleased with these analyses.
However, I still found some problems in the current MS.
Major problem
1, The authors used the genome, protein, cds data of Rosa rugosa, but did not cite the relevant article. I think the authors should not only cite this article but also talk about the importance of this article.
Minor problem
1, line 146, what are "pIs"? full name?
2, Usually, the phylogenetic tree was constructed using full length genes or proteins, but the authors only used seqs of domains, why? please provide evidences? Besides, NJ tree is not suggested but I suggest ML tree in phylogenetic analyses.
3, How did the authors normalize the FPKM, please provide methods!
Author Response
This MS reports the whole-genome wide analyses on the LBD genes in Rosa rugosa. The authors performed structural and expressional analyses. I found that these analyses are very comprehensive and informative. I am pleased with these analyses.
However, I still found some problems in the current MS.
Major problem
1, The authors used the genome, protein, cds data of Rosa rugosa, but did not cite the relevant article. I think the authors should not only cite this article but also talk about the importance of this article.
Answer: In the abstract part (L 21-25), the benefits of whole genome sequencing in recent years and the importance of rose genome are added, and the citation of Rosa rugosa genome article is added in the result part (line 107).
Minor problem
1, line 146, what are "pIs"? full name?
Answer: ‘pI’ is the abbreviation of ‘isoelectric point’ and we have revised it.
2, Usually, the phylogenetic tree was constructed using full length genes or proteins, but the authors only used seqs of domains, why? please provide evidences? Besides, NJ tree is not suggested but I suggest ML tree in phylogenetic analyses.
Answer: The phylogenetic tree was constructed based on the full protein sequences and the wrong statement has been modified (L159 and Part2.2). The LBD proteins were short sequences with few information sites, so the phylogenetic tree constructed by NJ method with few assumptions is relatively accurate. Besides, we constructed the ML tree and its topological structure is accordant with NJ tree.
3, How did the authors normalize the FPKM, please provide methods!
Answer: The normalization was the log base 2 of FPKM. We have added this in the legend of Figure 5.
Reviewer 2 Report
Authors successfully identified 41 LBD transcription factors of Rosa Rugosa and satisfactorily described their gene structure and cis-acting elements contained in their promoters. Nevertheless, I consider that gene expression analysis must be improved (both RNA-seq and qPCR). The salt stress experiment should be better described. Moreover, data presentation can be ameliorated to effectively support the role of RrLBD39, RrLBD40 and RrLBD25 as regulators of salt stress signaling.
Furthermore, some parts of introduction section are poorly written and must be revised, the legends of figure 2 and figure 5 contain some slight inaccuracies and the text in the figure 2 and figure 3 could be enlarged.
Author Response
Authors successfully identified 41 LBD transcription factors of Rosa Rugosa and satisfactorily described their gene structure and cis-acting elements contained in their promoters. Nevertheless, I consider that gene expression analysis must be improved (both RNA-seq and qPCR). The salt stress experiment should be better described. Moreover, data presentation can be ameliorated to effectively support the role of RrLBD39, RrLBD40 and RrLBD25 as regulators of salt stress signaling.
Answer: We have improved expression analysis in subheading 3.5 and supplied the details of the salt stress experiment in subheading 2.5 to support the salt responsive RrLBD candidates
Furthermore, some parts of introduction section are poorly written and must be revised, the legends of figure 2 and figure 5 contain some slight inaccuracies and the text in the figure 2 and figure 3 could be enlarged.
Answer: We have revised the introduction with more background information. The inaccuracies of legends and figures have been revised.
Reviewer 3 Report
This paper identified and phylogenetic analyzed LBD genes from the R. rugosa genome. Th present study provides some information for the evolution and development of LBD family in R. rugosa and indicates candidate genes involved in salt stress. But the paper needs improvement before acceptance for publication.
My detailed comments are as follows:
- The author stated that seven groups of LBD family, Ia, Ib, Ic, Id, Ie of Class I and IIa, IIb of Class II, were divided based on the sub-topological structure. However, Ic, Id, Ie seem to be same Class of IIa and IIb according to the the results showed in Figure 1. The author could revised phylogenetic analysis together with LBD family members in other species form published paper.
- The author should consistently rename the duplicated member of LBD family based on the phylogenetic analysis (Such as LBD12a and LBD12b, not as LBD16.1 and LBD16.2).
- The quality of English needs improving.
Round 2
Reviewer 2 Report
I urge the authors to involve an English native speaker in the manuscript revision or to use an English editing service to assist in bringing the English grammar and other English language issues to an acceptable level. There are several grammatical errors and sentence construction mistakes that make manuscript difficult to understand. (For example: lines 20-21, 35-36, 68-69, 71-76, 84-87, 88-91 etc.).
Moreover, the salt stress experiment was not still well described: plant growth conditions, pots dimensions, temperature, humidity, greenhouse or growth chamber etc. These aspects are to be clarified.
The legend of figure 2 still contains mistakes: UTR were highlighted in yellow and not in blue and CDS were highlighted in green and not in yellow.
Author Response
I urge the authors to involve an English native speaker in the manuscript revision or to use an English editing service to assist in bringing the English grammar and other English language issues to an acceptable level. There are several grammatical errors and sentence construction mistakes that make manuscript difficult to understand. (For example: lines 20-21, 35-36, 68-69, 71-76, 84-87, 88-91 etc.).
Moreover, the salt stress experiment was not still well described: plant growth conditions, pots dimensions, temperature, humidity, greenhouse or growth chamber etc. These aspects are to be clarified.
The legend of figure 2 still contains mistakes: UTR were highlighted in yellow and not in blue and CDS were highlighted in green and not in yellow.
Answer:
1.We have totally revised the languages and grammars according to the suggestions of English language editing company (ShineWrite.com).
2. We didn't describe the cultivation situation like pots dimensions, greenhouse or growth chamber because the rose bushes grow naturelly in undisturbed sands (not cultivation). In July 2019, the wild roses were digged out from its natural distribution area and the plants were treated by salt solution imidiatelly . The salt stress experiment was described with detail in subheading 2.5.
3. We have revised the mistakes in legend of figure 2.